# Low Uptake of the Second Dose of Human Papillomavirus Vaccine in Dar es Salaam, Tanzania

**DOI:** 10.3390/vaccines10111919

**Published:** 2022-11-13

**Authors:** Nchang’wa Nhumba, Bruno Sunguya

**Affiliations:** School of Public Health and Social Science, Muhimbili University of Health and Allied Sciences, Dar es Salaam 11103, Tanzania

**Keywords:** human papillomavirus, human papillomavirus vaccine, uptake of HPV vaccine second dose, Tanzania

## Abstract

Cervical cancer represents the most common neoplastic pathology among women, with a high burden of morbidity and mortality globally. Tanzania is no exception. The human papillomavirus (HPV) vaccine remains the most effective intervention to address such a burden. However, the uptake of the second dose to confer full immunity remains a challenge. This study aimed to assess the uptake and factors associated with the second dose of the HPV (HPV-2) vaccine uptake among adolescents in the Ilala municipality of Dar es Salaam, Tanzania. Using a quantitative cross-sectional study, data of 389 adolescent girls was collected using a self-administered structured questionnaire. Analyses were conducted using Statistical Package for the Social Sciences (SPSS) software through descriptive and multivariate logistic regression methods to determine uptake, characteristics, and factors associated with the uptake of the second dose of the HPV vaccine. Among the 389 adolescents, the uptake of the HPV-2 vaccine dose was only 21.3%, a lower level compared with the first dose of HPV vaccine (35.2%). Factors associated with the uptake of the HPV-2 vaccine were age (AOR 0.14, *p* = 0.008), positive attitude towards the HPV-2 vaccine (AOR 2.04, *p* = 0.023), and awareness of the HPV-2 vaccine (AOR: 9.16, *p* = 0.003). In conclusion, only one in five adolescents in the Ilala municipality received a second dose of HPV vaccine. Such low uptake was associated with attitude towards the HPV vaccine and low awareness of HPV-2 vaccines. Regular community sensitization and awareness campaigns by relevant authorities and implementers may help to increase the HPV vaccine uptake.

## 1. Introduction

Cervical cancer is one of the leading causes of cancer death among women and the fourth most frequently occurring malignancy among women globally [1,2]. It has an incidence rate of 530,000 cases and causes about 270,000 deaths annually, the highest being in eastern, western, and southern Africa, with age-standardized rates of 34.5, 33.7, and 26.8 cases per 100,000 people, respectively [3]. About eight in every ten cervical cancer deaths occur in low- and middle-income countries (LMICs) [4]. 

East Africa has the highest age-standardized incidence rate for cervical cancer at 42.7 per 100,000 women per year. Evidence suggests that Uganda harbors an estimated 33.6% of women with human papillomavirus (HPV) in the general population with 44 per 100,000 women developing the disease yearly [5]. In Tanzania, cervical cancer has remained the leading female cancer with a striking 9772 new cases costing 6695 lives annually [6]. 

Cervical cancer is associated with HPV infections, with serotypes 16 and 18 of HPV being responsible for about 70% of all invasive cervical cancer cases worldwide [7]. In sub-Saharan Africa, low knowledge and awareness of cervical cancer risk factors among the population and limited access to high-quality health-care services and cervical screening programs were vulnerable factors [8]. 

Vaccination against HPV is one of the major public health strategies for preventing cervical cancer in the general population. The vaccines are bivalent (CervarixTM) targeting HPV types 16 and 18, and quadrivalent (GardasilTM), targeting HPV 16 and 18 and HPV 6 and 11 that cause genital warts. Additionally, a nine-valent HPV vaccine (Gardasil 9TM) is available and targets HPV types 31, 33, 45, 52, and 58 in addition to HPV types 6, 11, 16, and 18 [9,10].

To ensure maximum protection, the HPV vaccination requires two doses within a six-month interval for adolescent girls aged 9–14 years as per the World Health Organization (WHO) recommendation [9]. Although in general the HPV vaccine coverage faces some challenges, the second dose has had a notably lower coverage compared to the first dose. Without duo doses, the complete prevention and protection against future cancers remains a challenge. [11]. 

Evidence from Tanzania indicates that only one in two adolescent girls aged 14 years were covered by the second dose of the HPV vaccine, while more than two in three received the first dose in 2019. Such vaccines are being given for free by the government of Tanzania. Factors for such low vaccine uptake remain unknown. Evidence on discrepancies pose a challenge for policy and planning to provide targeted interventions for cervical cancer prevention in Dar es Salaam and Tanzania as a whole. 

## 2. Methods

### 2.1. Study Area and Design

A cross-sectional quantitative study was conducted using self-administered questionnaires (Appendix A) in Ilala municipality, Dar es Salaam. The choice was based on evidence of the lowest coverage of the second dose of HPV vaccine compared to other municipalities.

#### Sampling and Sample Size

The required sample size of 388 respondents including a 5% non-response rate was determined using a formula given by Fisher et al. (2003). A three-stage sampling procedure for sampling school girls was applied in the study. At the first sage, three wards (Western Upanga, Gerezani, and Segerea) were selected in the Ilala municipality by simple random selection whereby the sampling frame was a list of all wards in the Ilala municipality. At the second stage, two secondary schools from each pre-selected ward were randomly selected, making a total of six secondary schools which were involved in the study (Tambaza, Jangwani, Dar es Salaam, Benjamin Mkapa, Migombani, and Ugombolwa secondary schools). A list of all secondary schools in the pre-selected wards formed a sampling frame. At the third stage, adolescent girls were randomly selected from their respective classes for participation in the study. This was carried out for each pre-selected secondary school until the required number of respondents was realized, with the attendance registers of the students forming the sampling frame.

### 2.2. Study Population

The study enrolled all adolescent girls aged 12–14 years in the pre-selected secondary schools. This is the age range group of adolescent girls that were thought to be in secondary schools, taking into account that some of them start schools earlier, especially in urban regions such as Dar es Salam.

### 2.3. Measurements

The dependent variable was uptake of the second dose of the HPV vaccine. Independent variables included factors such as socio-demographic characteristics, knowledge about the HPV-2 vaccine, perception, attitude towards the HPV vaccine, awareness on the second dose of the HPV vaccine and household wealth index. Knowledge of the respondents about the HPV vaccine was measured using fifteen questions, each with the response of “yes” or “no”, and the correct response scored one point. A score of more than or equal to 7 out of 15 points were considered good knowledge. Respondents who heard about the HPV-2 vaccine were considered as being aware of the second dose of the HPV vaccine [12]. Adolescents’ perception on the HPV vaccine was assessed with ten questions, each with a response of “yes”, “no”, or “don’t know”. A correct response to each question scored one point, and respondents who scored more than 50% were considered as having good perception [13]. Attitude was measured with ten questions to assess participants’ attitude on HPV vaccine series, the mean level of agreement based on a Likert scale from 5 (strongly agree) to 1 (strongly disagree) and Cronbach’s α was used to test for reliability, whereby a score of more than 0.5 was considered reliable [14]. Those who responded strongly agree and agree were considered as having a positive attitude while those responded strongly disagree, disagree, and undecided were considered as having a negative attitude. The questions were adapted from Witte, Meyer, and Martell (2001). The variables for household wealth index were reduced using principal component analysis (PCA) to include those that determined the wealth index. Factor loadings were used to derive levels of wealth index and categorize into quintiles (wealthiest, fourth, middle, second, and the poorest).

### 2.4. Data Management and Analysis

Data was reviewed for completeness, and coded and analyzed using SPSS (Statistical Package for the Social Science) version 26. Summary statistics such as frequencies and percentages of different variables were computed, then univariate Logit mode was conducted to measure the association between explanatory variables and outcome variables. At multivariate analysis, all independent factors with *p* < 0.2 were included to obtain adjusted odds ratio (AOR) with 95% (CI) to test factors independently associated with an uptake of the second dose of the HPV vaccine. Categorical variables were summarized as percentages and frequencies and presented in tables and figures. 

### 2.5. Ethical Consideration

Ethical approval of the study was obtained from the Muhimbili University of Health and Allied Sciences (MUHAS) Research and Ethics Committee with an IRB number MUHAS-REC-05-2021-602 before conducting data collection. Permission to carry out the study in the region was obtained from the regional and district council. The purpose of the study was explained to participants and written informed consent was obtained from school adolescent girls aged 18 years and above before the interview was conducted. Heads of selected secondary schools consented on behalf of the girls aged less than 18 years who could not consent. Information provided by the participants was disclosed only for research purposes.

## 3. Results

### 3.1. HPV Vaccine Uptake

Out of 389 respondents, only 83 (21.3%) had received the second dose of HPV vaccine, 137 (35.2%) had received the first dose of HPV vaccine shot which was high compared to the second shot. The majority of the respondents had not received both first and second doses of the HPV vaccine (Figure 1). Therefore, the uptake level of the second dose of the HPV vaccine was lower compared with the first dose in the Ilala municipality, Dar es salaam.

### 3.2. Socio-Demographic Characteristics of HPV-2 Vaccine Uptake

A total of 57 (14.7%) adolescents were between 12 and 13 years, and of them, only 2 (3.5%) had received a second dose of the HPV vaccine. Of the older adolescents, 81 (24.4%) had received the second dose. Adolescents whose parents had no formal education were less likely to receive a second HPV vaccine (*p* < 0.001) compared to those whose caregivers had secondary education. A total of 53 (24.7%) adolescents whose parents were self-employed received a second dose, a proportion higher than those whose parents were either employed or unemployed (*p* = 0.032). There was no significant difference in second dose uptake with regards to household wealth level. (Table 1).

### 3.3. Factors Associated with Uptake of the Second Dose of HPV Vaccine

After adjusting for confounders, only two factors (attitude and awareness) were found to be independently associated with second HPV vaccine uptake with *p*-value < 0.05. Other factors were not statistically significant and therefore did not fit to be independent predictors of second dose of HPV vaccine.

A total of 219 (56.3%) respondents with a positive attitude towards HPV vaccine were positively associated with the uptake of HPV-2 (AOR: 2.04; 95% CI: 1.10–3.76), *p*-value = 0.023, and were 2.04 times more likely to receive the HPV-2 vaccine compared to those with a negative attitude.

A total of 322 (82.8%) respondents with awareness about HPV-2 vaccine were 9.16 times more likely to receive a second HPV vaccine shot compared to the group without awareness (AOR: 9.16; 95% CI: 2.11–39.85), *p*-value = 0.003. Therefore, positive attitude and awareness on HPV-2 vaccine were independent predictors of HPV-2 uptake. 

A total of 147 (37.8%) respondents had good perception about the HPV vaccine, and such respondents were 1.58 times more likely to receive the HPV-2 vaccine compared to those with poor perception (AOR: 1.58; 95% CI: 0.88–2.84) *p*-value-0.124. Those with good knowledge of the HPV vaccine were 186 in total (48.2%), and were 1.05 times more likely to receive the HPV-2 vaccine compared to those with poor knowledge (AOR: 1.05; 95% CI: 0.57–1.95) with *p*-value 0.878 (Table 2). However, knowledge and perception did not achieve significant levels of being independent predictors of the second dose of the HPV vaccine (HPV-2) uptake as their *p*-values were more than 0.05.

## 4. Discussion

The study examined the uptake of the second dose of HPV vaccine and factors that may be associated with its uptake in Dar es salaam, Tanzania. The study findings provide recommendations to policy makers and other stakeholders to improve strategies for increasing the second dose coverage, to provide full protection to women from cervical cancer.

In this study, findings revealed that the uptake of the second dose of the HPV vaccine was lower compared to the first dose. Age, attitude, and awareness were independent predictors of HPV vaccine second-dose uptake.

Only one in five adolescent girls (21.3%) had taken a second dose of the HPV vaccine in Dar es salaam, Tanzania. This is a lower uptake compared to the reported 35.2% for the first dose. This finding suggests a low uptake trend, similar to another study in Uganda which revealed 17.6% of adolescents had taken the second HPV vaccine dose [15]. The low uptake of the second HPV vaccine shot may be due to lack of awareness on the importance of the second HPV vaccine dose of HPV, similar to other contexts in the US [16]. Previous evidence has demonstrated a high uptake of the second HPV vaccine dose in developed countries compared to the developing countries. However, the uptake of first dose is consistently high compared to the second dose for both high and low middle-income countries [17]. Lack of access to health facilities, socio-economic status, cultural beliefs, and lack of awareness and knowledge about HPV vaccine among adolescents and the community at large could explain such differences.

Age was associated with an uptake of the second dose of HPV vaccine in this study. Adolescent girls aged between 12 and 13 years were 86% less likely to receive the second HPV vaccine shot compared to their elders, similar to other contexts [18,19]. In addition to higher perception among older adolescents, their younger counterparts may still feel and consider themselves passive in making decisions and would be happy for their parents to decide for them [20]. 

Attitude was significantly associated with an uptake of the second dose of the HPV vaccine in this study. Adolescent girls with positive attitude towards HPV vaccine were positively associated with the second dose of the HPV vaccine uptake, similar to other studies from low- and high-income countries [14,21,22,23].

Awareness about the HPV vaccine second dose among school adolescent girls in this study was also found to be an independent predictor of second HPV-vaccine uptake. Public awareness programs on HPV vaccination are of paramount in increasing the second dose of HPV vaccination among adolescent girls [11]. This finding correlates with studies carried out in both developing and developed countries [15,24]. Although the majority of the respondents were aware of the HPV vaccine, the proportion was lower compared to a similar study done in Hong Kong [19].

Perception on the HPV vaccine second dose in this study was not an independent predictor of uptake. Comparatively, other studies in particular showed no significant association between perception and second dose of HPV vaccine uptake [25]. Contrary to this study’s findings, other studies have shown perception to be negatively associated with uptake of the second dose of HPV vaccine: that the vaccine is unnecessary and acts as a permitting signal to adolescent girls to have sex [26]. The reason for this difference could be due to mistrust, low level of perceived benefit, or a negative belief of the vaccines amongst adolescent girls and parents [27].

HPV vaccine knowledge promotes coverage in various contexts. Media, medical practitioners, teachers, and parents play a significant role in fostering HPV vaccine knowledge which in turn leads to increased HPV-2 vaccination coverage [28]. In this study, knowledge on the HPV vaccine among school girls was found to be low, but not significantly associated with HPV 2 vaccine uptake. This finding was similar to another study carried out in Brazil [29]. The reasons are due to lack of perceived benefits and the importance of the HPV vaccine to adolescent girls [30].

The current study showed that the socio-economic status of individual adolescent girls was not significantly associated with the uptake of the second HPV vaccine. This was contrary to other studies which showed an association of HPV vaccine uptake with lower economic status and middle-wealthy quantiles [31,32]. This could be explained by the fact that the HPV vaccine was introduced in Tanzania in 2018, one year before the study was conducted, making it relatively new to the community and with a low perceived benefit of the vaccine from adolescent girls and parents [33]. Similarly, the association between parental educational level and occupation with the uptake of the second dose of the HPV vaccine did not reach a statistically significant level similar to other studies [34].

The evidence presented in this study is not short of limitations. The study assessed level of uptake for the second dose but the context may not be the same as in the first dose owing to massive campaigns, awareness, and communications made to adolescents and their caregivers. The study design was cross-sectional, which has its own limits in making causal inferences between the main outcome and independent variables.

In conclusion, this study showed that the uptake of second dose of the HPV vaccine among adolescent girls in the Ilala municipality was low. Factors associated with the second dose of the HPV vaccine uptake included good attitude towards the HPV vaccine and awareness about HPV vaccines. This evidence calls for a regular HPV vaccination community sensitization about HPV and its vaccine to increase awareness, and therefore improve attitudes towards vaccination itself.

## Figures and Tables

**Figure 1 vaccines-10-01919-f001:**
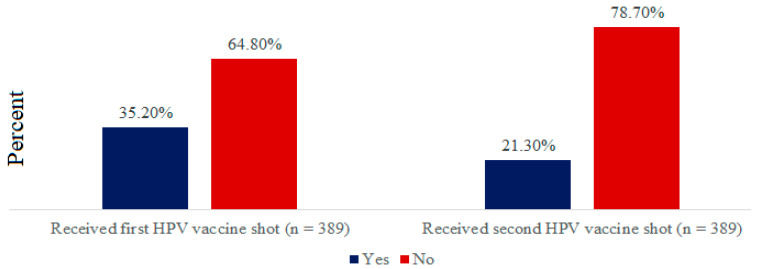
Uptake of the first and second dose of HPV vaccine.

**Table 1 vaccines-10-01919-t001:** **Characteristics of HPV-2 uptake**.

Variables			Received Second Dose of HPV Vaccine		
	Yes		No		*p*-Value (χ2)
	N	(%)	N	(%)	
Age (years)					
12–13	2	(3.5)	55	(96.5)	<0.001
14+	81	(24.4)	251	(75.6)	
Parental status					
One parent or orphan	14	(20.9)	53	(79.1)	0.923
Two parents	69	(21.4)	253	(78.6)	
Parental educational level					
No formal or primary education	15	(19.0)	64	(81.0)	<0.001
secondary or above	68	(21.9)	242	(78.1)	
Parents’ occupation status					
Employed	20	(14.3)	120	(85.7)	0.032
Self employed	53	(24.7)	162	(75.3)	
Unemployed	10	(29.4)	24	(70.6)	
Household Wealth Index					
Lowest	19	(24.4)	59	(75.6)	0.122
Second	21	(27.3)	56	(72.7)	
Middle	20	(25.6)	58	(74.4)	
Fourth	10	(12.8)	68	(87.2)	
Highest	13	(16.7)	65	(83.3)	
Total	83	(21.3)	306	(78.7)	

**Table 2 vaccines-10-01919-t002:** **Logistic regression for the factors associated with HPV-2 uptake**.

Variables	N (%)	Un Adjusted OR (95%CI)	*p* Value	Adjusted OR (95%CI)	*p* Value
*Age (years)*					
12–13	147 (37.8)	8.87 (2.12 37.19)	0.003	0.14 (0.03 0.59)	0.008
14+	332 (85.3)	1		1	
*Parental educational status*					
No formal or primary education	79 (20.3)	1.09 (0.59 2.00)	0.79	0.61 (0.30 1.22)	0.159
Secondary or above	310 (79.7)	1		1	
*Parental occupational status*					
Employed	140 (36.0)	2.50 (1.04 6.01)	0.04	0.39 (0.14 1.05)	0.063
Self employed	215 (55.7)	1.27 (0.57 2.84)	0.55	0.70 (0.28 1.73)	0.444
Unemployed	34 (8.7)	1		1	
*Household wealth index*					
Lowest	78 (20.1)	1.61 (0.73 3.54)	0.132	1.39 (0.57 3.38)	0.468
Second	77 (19.8)	1.87 (0.86 4.08)	0.236	2.07 (0.87 4.88)	0.098
Middle	78 (20.1)	1.72 (0.79 3.77)	0.114	1.52 (0.65 3.53)	0.330
Fourth	78 (20.1)	0.73 (0.30 1.79)	0.173	0.57 (0.22 1.48)	0.251
Highest	78 (20.1)	1		1	
*Perception of HPV-2 vaccine*					
Good perception	147 (37.8)	1.84 (1.13–3.01)	0.015	1.58 (0.88–2.84)	0.124
Poor perception	242 (62.2)	1		1	
*Attitude towards HPV-2 vaccine*					
Positive attitude	219 (56.3)	2.24 (1.33–3.79)	0.003	2.04 (1.10–3.76)	0.023
Negative attitude	170 (43.7)	1		1	
*Awareness of HPV-2 vaccine*					
Heard about HPV vaccine	322 (82.8)	10.92 (2.62–45.62)	0.001	9.16 (2.11–39.85)	0.003
Not heard about HPV vaccine	67 (17.2)	1		1	
*Knowledge of HPV-2 vaccine*					
Good knowledge	186 (48.2)	1.67 (1.02–2.73)	0.041	1.05 (0.56–1.95)	0.878
Poor knowledge	203 (52.2)	1		1	

## Data Availability

Not applicable.

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
