# Peer review of "Low Uptake of the Second Dose of Human Papillomavirus Vaccine in Dar es Salaam, Tanzania"

_vaccines, 2022, doi:10.3390/vaccines10111919_

Round 1

Reviewer 1 Report

Questionary should be included in the MS

---updated detailed report

The MS itself is quite interesting. However I may adress the following comments to include in my evaluation: The introduction part could be improved. The social aspects of the vaccine system should be included. The cost of the vaccine and if it is partially or totally covered by the public health system.  The questionary should be included to evaluate the accuracy of the results. It is not clear the methodology when related the ammount of questions, all the girls answered all of them?  The discussion should be improved also when regarding the possibilities that arise from the results found. It is clear that cannot be demostrated but the authors should propose some. How is the general knowledge of vaccines in the society? How is the vaccinal coberture for others infections? Is here different? What strategy Could improve the results? Girls of 12 years are prepared to know and evaluate the importance of the vaccine decision?

Author Response

COVER LETTER WITH RESPONSE TO REVIEWER  1 COMMENTS.

This is the cover letter herewith the table that summarises the detailled changes for the editors’ and referees’ approval.

Manuscript title: LOW UPTAKE OF THE SECOND DOSE OF HUMAN PAPILLOMAVIRUS VACCINE IN DAR ES SALAAM, TANZANIA.

Author: NCHANG’WA NHUMBA

S/N

Manuscripts’ Section

Comments from the Reviewer

Authors’ response

Page No. where changes have been made

1

 Manuscript title

2

Abstract

3

Keywords

4

Introduction

The cost of the vaccine and if it is partially or totally covered by the public health system

The introductiona part could be improved

Thanks for such comment, the vaccine is given for free by the Government

Thanks, taken care.

2

Page 1

5

Methods

6

Results

What would be the strategies for improving the vaccine uptake in Tanzania and similar countries?

According to results obtained, factors like awareness and good altitude were found to be independently associated with increased uptake of the vaccine. Therefore, increasing awareness to the community on the importance of this vaccine to adolescent girls and changing peoples’ bad atitude towards HPV vaccine will increasing the vaccination uptake.

Page 7, on the conclusion part.

7

Discussion

The discussion should be improved

Has been addressed, thank you!

7

8

Conclusion

9

References

10

General comments

How is the vaccinal coberture for others infections? Is here different?

How is the general knowledge of vaccines in the society?

The questionary should be included to evaluate the accuracy of the results

The vaccinal corbeture is different for other infections. This could be due to recently introduction of HPV vaccine in the country.The vaccine itself and the disease it prevents is also believed to be one among the factors promoting sexual prostitute to young girls.

The general knowledge is inadequate to the society especially for HPV vaccine.

Already included

Appendix 1

Reviewer 2 Report

Well written.

It is not clear whetherbthe vaccine is given freebby the Government or it is available for purchase only. Please clarify.

It is not clear how the households were classified into economic groups. Please clarify. Is the sample represntative of the country's economic groups? Please clarify.

Can you supplement or substitute the tables with graphs or figures? This will render it easier to understand. 

Would the sample be representative of the whole country? How would it represent?

Is there a rural/urban divide in Tanzania?

What would be the solutions for increasing the vaccine uptake in Tanzania and similar countries? Would a single dose be sufficient?

How many girls fall outside the 9 to 14 year group in the districts or schools covered? Do older girls have better adherence?

A few spelling mistakes need correction tructuted for structured.

Author Response

COVER LETTER WITH RESPONSE TO REVIEWER 2 COMMENTS

This is the cover letter herewith the table that summarises the detailled changes according to second reviewers’ comments.

Manuscript title: LOW UPTAKE OF THE SECOND DOSE OF HUMAN PAPILLOMAVIRUS VACCINE IN DAR ES SALAAM, TANZANIA.

Author: NCHANG’WA NHUMBA

S/N

Manuscripts’ Section

Comments from the Reviewer

Authors’ response

Page No. where changes have been made

1

 Manuscript title

2

Abstract

Spelling mistakes, tructured for structured

Thanks for such observation,

It’s corrected

1

3

Keywords

4

Introduction

Not clear whether the vaccine is given free by the Government.

Thanks for such comment, the vaccine is given for free by the Government

2

5

Methods

Not clear how the households were classified into economic group

Would the sample be representative of the whole country? How would it represent?

Noted, and has been addressed.

Thank you so much.

The study design was a cross-sectional that limits in making causal inference between outcome  and independent variable

3

6

Results

What would be the solutions for increasing the vaccine uptake in Tanzania and similar countries?

Can you supplement or substitute the tables with graphs or figures? This will render it easier to understand

According to results obtained, factors like awareness and good altitude were found to be independently associated with increased uptake of the vaccine. Therefore, increasing awareness to the community on the importance of this vaccine to youg girls and changing peoples’ bad atitude towards HPV vaccine will increasing the vaccine uptake.

Yes, suplemented with a table for uptake level of HPV vaccines.

Page 7, on the conclusion part.

Page 4

7

Discussion

8

Conclusion

9

References

10

General comments

Is there a rural/urban divide in Tanzania

Would a single dose be sufficient?

Yes, it is!

Our study was conducted in urban region Ilala, Dar es salaam city

Single dose is not sufficient according to WHO recommendation, though some studies have shown that a single dose can as well be sufficient for prevention of cervical cancer. Still under study

Page 1, On introductory part, second last paragraph